# Diagnostic Dilemma of Recurrent Pulmonary Embolism

**DOI:** 10.3390/diagnostics10020096

**Published:** 2020-02-11

**Authors:** Alexandra Dadarlat-Pop, Irina Burian, Laura Cadis, Raluca Tomoaia, Alexandru Oprea

**Affiliations:** 1Cardiology Department, Heart Institute Niculae Stăncioiu, 19-21 Moților street, 400001 Cluj-Napoca, Romania; dadarlat.alexandra@yahoo.ro (A.D.-P.); irinaburian@yahoo.com (I.B.); laura30@yahoo.com (L.C.); alexandru_oprea2002@yahoo.com (A.O.); 2Department of Cardiology, Iuliu Haţieganu University of Medicine and Pharmacy, 8 Victor Babes street, 400012 Cluj-Napoca, Romania; 3Cardiovascular Surgery Department, Heart Institute Niculae Stăncioiu, 19-21 Moților street, 400001 Cluj-Napoca, Romania

**Keywords:** popliteal venous aneurysm, recurrent pulmonary embolism, therapy

## Abstract

Popliteal venous aneurysms are rare vascular disorders associated with a high risk of pulmonary embolism. We present the case of a 56-year-old woman hospitalized for a third episode of unprovoked pulmonary embolism. Venous ultrasonography identified a popliteal aneurysm, repeatedly missed by two-point compression venous ultrasonography, which was eventually confirmed by a magnetic resonance examination. Because of its highly symptomatic nature despite optimal anticoagulant treatment, the decision was made to undergo surgery, consisting of aneurysmectomy followed by patch angioplasty. The goal of this paper is to report a rare case of popliteal venous aneurysm and its treatment strategies and postoperative evolution.

## 1. Introduction

Popliteal venous aneurysms (PVAs) are rare pathologic vascular disorders. They may have various aetiologies including congenital, trauma, varicose veins, localized degenerative changes, and inflammation. Moreover, they usually remain silent until a thromboembolic event is developed. Given the rarity and potential fatal complications of this condition, without an established consensus of its management we report a case of primary popliteal venous aneurysm and discuss the therapeutic approach.

## 2. Patient Information and Physical Examination

A 56-year-old female patient presented to the emergency room with a recent syncope followed by dyspnoea of acute onset. The medical history was significant for a left Baker’s cyst, Hashimoto thyroiditis, and recurrent episodes of unprovoked pulmonary embolism (2017, 2018). During previous hospitalizations, Doppler venous ultrasonography showed no signs of deep vein thrombosis. Furthermore, no predisposing factors were identified, and the patient underwent screening tests for malignancy and blood tests for acquired or inherited coagulation disorders, all of which had been negative.

Since the hospital discharge after the first diagnosis of pulmonary embolism, the patient was commenced on long-term treatment with direct oral anticoagulant which she has been following up until the current hospital admission.

## 3. Clinical Examination

Physical examination revealed a hemodynamically stable patient with an oxygen saturation of 96% without additional oxygen, blood pressure of 110/70 mmHg, heart rate of 76 bpm, respiratory frequency (RF) of 18 r.p.m., and a temperature of 36.8 °C. Heart sounds were regular, and no cardiac murmurs were detected. Lung sounds were normal, with no dry or moist rales. Examination of the lower extremities was unremarkable—no palpable masses and no signs of deep vein thrombosis were identified. By using the recommended pulmonary embolism (PE) prediction rules, the revised Geneva and the Wells score, the patient was classified in the moderate-probability category, thus having a 20–30% risk of PE. The pulmonary embolism severity score (PESI) was 56, corresponding to a low-risk.

## 4. Diagnostic Assessment

Blood work showed elevated plasma D-dimer levels (2125 ng/mL) and normal cardiac biomarkers.

ECG showed signs of RV strain, with inverted T waves in leads V1–V4, and incomplete right bundle-branch block.

Echocardiogram revealed no signs of RV pressure overload or dysfunction.

A computed tomographic pulmonary angiography was performed that revealed filling defects located in the segmental arterial branches of the right inferior lobe, confirming the diagnosis of pulmonary embolism. 

Two-point compression ultrasonography showed no signs of deep vein thrombosis. However, a complete ultrasonography of the lower extremity venous system in real-time B-mode revealed a 6 × 5 × 4 cm anechoic mass located in the cranial extremity of the left popliteal fossa, contiguous to the popliteal vein, with sluggish, swirling blood flow, but no signs of thrombosis (Figure 1). 

Doppler spectral analysis revealed low velocity blood flow with normal phasic variation corresponding to a venous waveform. 

The ultrasonography also identified the Baker’s cyst as a fluid-filled “speech bubble” structure at the postero-medial knee with a “neck” at its deepest extent.

Knee MRI was performed to better describe the extension and anatomy of the aneurysm. It described an aneurysmal dilatation of the distal popliteal vein measuring 64 × 19 × 31 mm, with turbulent blood flow which exerted a mass effect on the adjacent muscle groups (Figure 2).

## 5. Treatment

After a Heart Team session, and in accordance to the patient’s wishes, the decision was made to proceed with surgical repair of the aneurysm.

### Surgery

Access to the popliteal fossa was obtained via a posterior approach.

Intraoperative findings confirmed the presence of 6/4 saccular aneurysm located along the distal popliteal vein (Figure 3).

The popliteal vein and the aneurysm were isolated, followed by longitudinal incision and drainage of the aneurysm, which revealed no thrombotic material. Next, the surgical procedure consisted of tangential aneurysmectomy followed by patch angioplasty with bovine pericardium (Figure 4).

Microscopic examination with tricrom masson staining protocol showed continuous thinned venous layer with fragmentation of elastin fibres (blue arrow) and few smooth muscle fibers (white arrow) confirming the diagnosis of popliteal venous aneurysm (Figure 5).

Early post-surgery, the patient developed a hematoma that required surgical haemostasis and drainage. Otherwise, the recovery was uneventful. Post-discharge medication consisted of oral anticoagulant and aspirin.

## 6. Follow-up

The patient remained symptom-free at 2 months after the surgery. The antiplatelet treatment was discontinued after 3 months. After consulting with the patient, the decision has been made to maintain the anticoagulant treatment.

## 7. Discussion

Venous aneurysms are rare findings, having different aetiologies and locations throughout the body [1,2]. They are described in various anatomical locations as head and neck, thoracic, intraabdominal, or extremities, some of them being associated with congenital malformations such as Klippel-Trenaunay syndrome [1]. As for arterial aneurysms, a distinction is made between fusiform and saccular aneurysms, primary and secondary venous aneurysm, respectively. While the aetiology of primary venous aneurysms remains unclear, venous hypertension, external compression, or direct trauma are the most frequent causes of secondary venous aneurysms. Primary popliteal venous aneurysms (PVA) are the most common deep venous aneurysms of the lower extremity. They were first described by Dahl et al. in 1976 [2]. They present several histological changes, such as decreased smooth muscle cells in the media and a fibrous intima [1]. The major complications of these aneurysms are thromboembolic events. Unfortunately, based on the existing data, between 24% and 51% of patients with popliteal venous aneurysm have pulmonary embolism (PE) as the presenting symptom, which may be a life-threatening condition [2]. Actually, large aneurysms are associated with a 70–80% chance of PE formation despite efficient anticoagulation [2]. PE clinical signs and symptoms are non-specific, including dyspnoea, chest pain, haemoptysis, or syncope [3]. When syncope is a symptom of PE, it is associated with a higher risk of sudden death [4]. On the other hand, patients with deep venous thrombosis may frequently develop silent recurrent PE. Other clinical presentations may be related to chronic venous disease or local mass effect. However, the risk of rupture is low [5]. Approximately two-thirds of PVAs are saccular and the majority are located in the left lower limb because of the compression of the left common iliac vein by the right common iliac artery [6].

Also, a higher prevalence in women was reported. A possible explanation could be the oestrogen-related effects on angiogenesis [5].

### 7.1. Diagnosis

Ultrasound, CT, or magnetic resonance (MR) imaging are used as non-invasive diagnostic tools for this condition. We believe that the tool of choice remains the color duplex ultrasound, which besides describing the morphology of the aneurysm can predict the risk of thrombus formation by detecting the turbulent flow in the aneurysmal segments. But, when an intervention is warranted, computed tomography venography or magnetic resonance venography may be required for preoperative planning. Diagnosing PVAs venous thromboembolic complications should follow the guideline recommended diagnostic workup [3]. In our case, using the guideline proposed algorithm for diagnosing PE, by combining the pre-test prediction rules—the Wells score or the modified Geneva score with D-Dimers determination correctly lead to the indication of performing imaging testing, which confirmed PE.

### 7.2. Surgical Management

Currently, there are no criteria with regard to the venous size in defining venous aneurysms. Isolated dilatation of one and a half, two or three times more than the native vein have been described in the existing literature [1,2,4]. Moreover, no consensus regarding the management of primary venous aneurysms exists. 

Even though there is controversial data regarding the therapeutic management of PVAs, expectant monitoring, medical management with anticoagulation, and surgical repair are the possible therapeutic plans [1]. Anticoagulation may reduce the risk of thromboembolism but may theoretically worsen the outcome in the event of aneurysmal rupture [5]. The natural history of asymptomatic PVAs remains unknown, therefore the ideal treatment strategy is controversial. But, asymptomatic saccular or large fusiform PVAs are associated with potentially life-threatening embolic events, therefore surgical repair is generally advised. There are studies showing that a popliteal venous aneurysm with turbulent flow detected by duplex ultrasound and a diameter of more than 20 mm has a firm indication for surgery because of the unpredictable risk of PE [5]. Also, studies suggest surgery as the method of choice in patients with recurrent thrombosis or pulmonary embolism after anticoagulation or having a high risk of venous thrombosis [5]. Nasr et al. showed in a study that anticoagulation therapy alone fails in 43% of cases in preventing thromboembolic events in patients with primary venous aneurysms [7]. As in our case, patients may develop recurrent pulmonary emboli despite anticoagulation therapy when a nonoperative management strategy is employed.

The operative technique must be selected on a case-by-case basis. Posterior access is the most common, followed by resection of the aneurysm sac and lateral venorrhaphy to reconstruct the vein. Primary venorrhaphy can be done if the aneurysm wall appears to be of good integrity. In our case, intraoperative findings of a severely diseased venous wall dictated the use of a patch. The procedure usually has few complications, including transient common peroneal nerve palsy and hematoma formation, as in our case [8].

Whereas endovascular techniques have become a mainstay of vascular surgery, they currently have no defined role in the treatment of venous aneurysms [2].

### 7.3. Postoperative Management

There are cases of recurrent popliteal venous aneurysms after surgical patchplasty that require surgical reintervention. The role and duration of anticoagulation in postoperative asymptomatic patients is unclear. Several reports suggest oral anticoagulation between 3–12 months, the use of compression stocking, or lifelong aspirin therapy [1].

The small number of existing reported cases make definitive recommendations hardly to be made. Even though current guidelines for the diagnosis and management of acute pulmonary embolism [3] do not mention venous aneurysms of the extremities as major transient factors for pulmonary embolism we strongly believe that it is very important for physicians to suspect PVAs until it is otherwise ruled-out. In this context, we hope that our case adds evidence to the body of knowledge regarding primary popliteal venous aneurysms.

## Figures and Tables

**Figure 1 diagnostics-10-00096-f001:**
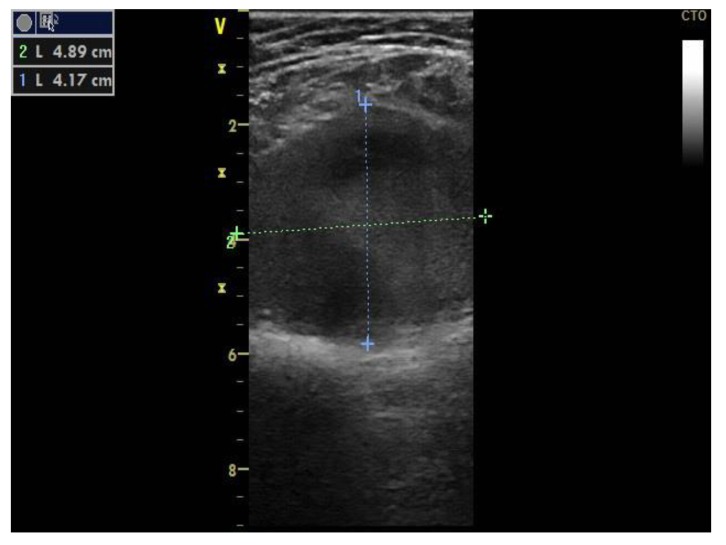
Proximal left popliteal vein transverse axis view demonstrates a simple saccular dilatation, with partial thrombosis. The dotted lines represent the PVA’s dimensions.

**Figure 2 diagnostics-10-00096-f002:**
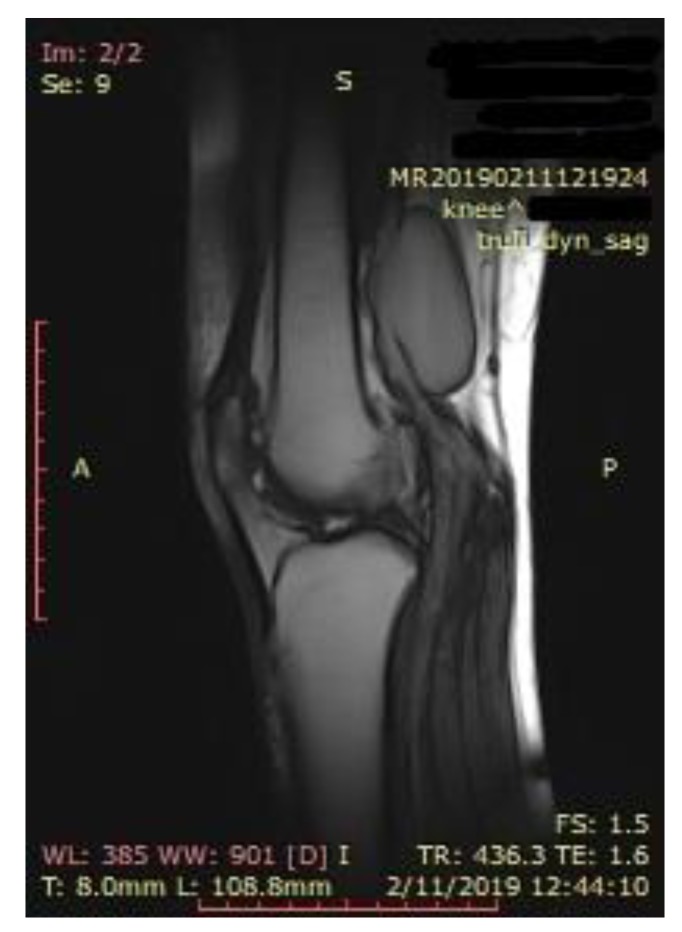
Magnetic resonance (MRI) of the knee, longitudinal axis view, showing the anatomy and location of the popliteal vein aneurysm.

**Figure 3 diagnostics-10-00096-f003:**
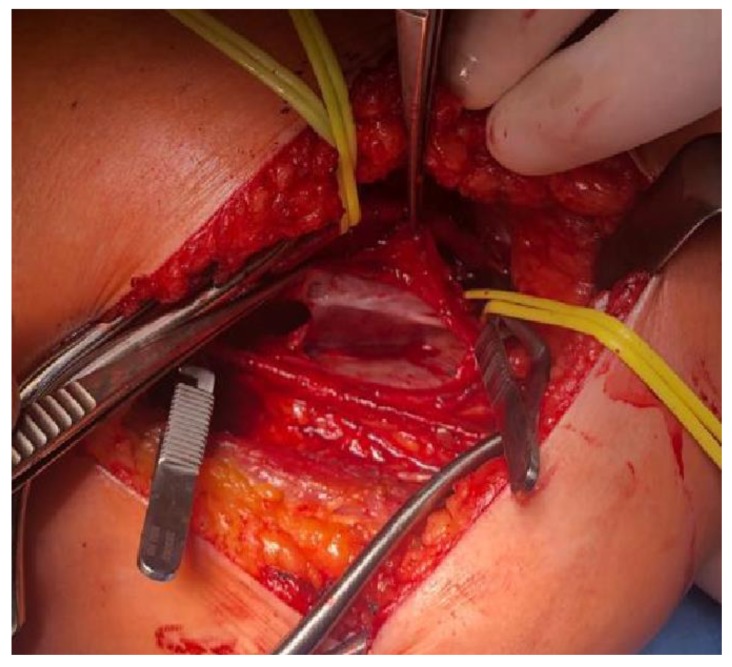
Intraoperative view after the isolation, incision and drainage of the popliteal vein aneurysm.

**Figure 4 diagnostics-10-00096-f004:**
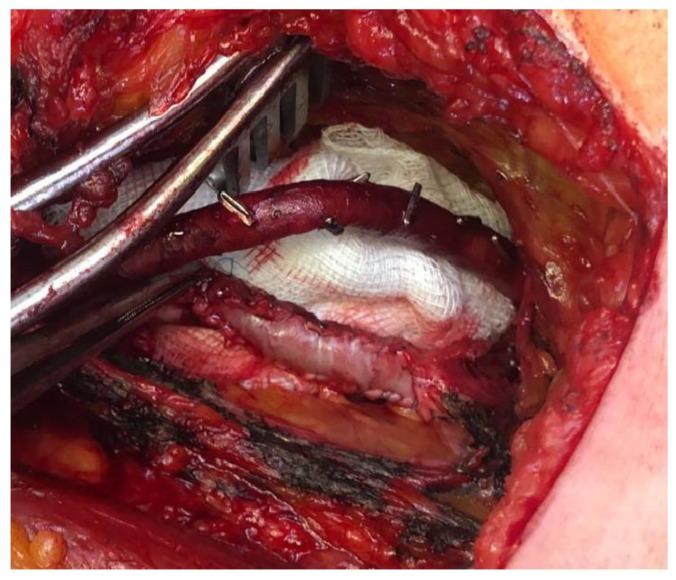
Tangential aneurysmectomy of the PVA and patch angioplasty with bovine pericardium.

**Figure 5 diagnostics-10-00096-f005:**
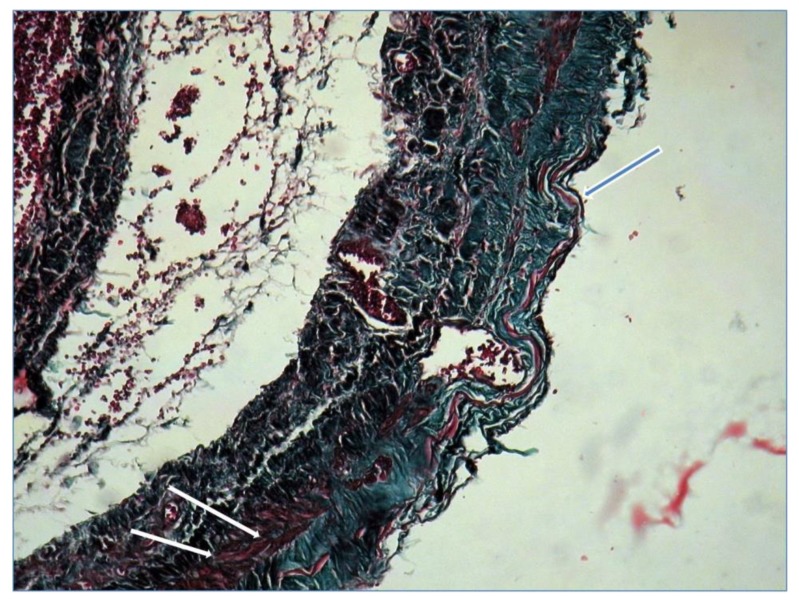
Microscopy confirming the diagnosis of popliteal venous aneurysm.

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
