# Peer review of "Diagnostic Dilemma of Recurrent Pulmonary Embolism"

_diagnostics, 2020, doi:10.3390/diagnostics10020096_

Round 1
Reviewer 1 Report
In this case-report entitled by Dadarlat-Pop et al., describing a case of venous aneurism of popliteal vein, Authors open the door to the discussion on the occurrence of pulmonary embolism as complication of this rare venous disease. The description of the case-report is interesting, but the Discussion on the potential occurrence of pulmonary embolism needs some improvement.
Specific Comments
Authors should mention the use of Wells and revised Geneva Scores as reference scores for the pre-test probability. What was the patient's score achieved? Both Wells and revised Geneva scores were born in the Emergency room, but have been recently tested even behind the Emergency Department, to assess their performance in a ward other than the emergency room (Di Marca S. et al. J Am Geriatrics Soc 2015). Taking this paper as starting-point, a comment on the usefulness of these scores is required. Syncope may be the first symptom of pulmonary embolism (Prandoni P. et al., N Engl J Med 2017), and this should be pointed out as risk of sudden death. It should be emphasized that patients with recurrent pulmonary embolism may be symtomless, or showing a symptom apparently misleading (i.e., syncope), driving the physician's attention toward the heart.Author Response
First of all, we sincerely thank you for the constructive criticism and valuable comments, which were of great help in revising our case report.
We responded to each comment individually:
Comment 1: Does the introduction provide sufficient background and include all relevant references?
Response: In the Introduction section we have made several modifications, all marked in yellow in the revised paper, adding information in line with the existing data.
Comment 1: The description of the case-report is interesting, but the Discussion on the potential occurrence of pulmonary embolism needs some improvement.
Response: We added some data regarding the rate of occurrence of PE in patients with asymptomatic venous aneurysms, as you can see in the Discussion section of the paper. Taking into account the relatively rare findings- PVAs there is not enough evidence in the literature regarding the natural history of asymptomatic PVAs and therefore the ideal treatment strategy remains controversial. But, the major complications of PVAs are the thrombo-embolic events. So, large venous aneurysms should undergo surgical repair in order to prevent life-threatening embolic events.
Comment 2: Authors should mention the use of Wells and revised Geneva Scores as reference scores for the pre-test probability. What was the patient's score achieved? Both Wells and revised Geneva scores were born in the Emergency room, but have been recently tested even behind the Emergency Department, to assess their performance in a ward other than the emergency room (Di Marca S. et al. J Am Geriatrics Soc 2015)
Response: The Wells original score was 4, the patient having an intermediate clinical probability and the simplified two-level Wells score predicted PE as “likely”. She had a 9-point modified Geneva score, which classified her in the moderate risk group for PE, thus having a 20-30% risk of PE. So, by assessing the clinical probability using the prediction scores and determining D-Dimer levels we had a high suspicion of PE, therefore we performed the pulmonary angio-scan which confirmed the diagnosis of PE. Therefore, we highlight the good diagnostic performance of both clinical prediction models in patients with PVAs and PE.
Comment 3: Syncope may be the first symptom of pulmonary embolism (Prandoni P. et al., N Engl J Med 2017), and this should be pointed out as risk of sudden death. It should be emphasized that patients with recurrent pulmonary embolism may be symptomless, or showing a symptom apparently misleading (i.e., syncope), driving the physician's attention toward the heart.
Response: Thank you for this valuable comment. We added the following data in the Discussion section, taking into account your recommendation: PE clinical signs and symptoms are non-specific, including dyspnea, chest pain, haemoptysis or syncope. When syncope is a symptom of PE, it is associated with a higher risk of sudden death (Paolo Prandoni, Anthonie W A Lensing, Martin H Prins, Maurizio Ciammaichella, Marica Perlati, Nicola Mumoli, et al. Prevalence of Pulmonary Embolism Among Patients Hospitalized for Syncope. N Engl J Med. 375 (16), 1524-1531). On the other hand, patients with deep venous thrombosis may frequently develop silent recurrent PE.
The changes in the text appear in yellow in the revised paper.
We sincerely consider your comments as being of great help in revising the paper.
Thank you!

Reviewer 2 Report
This an interesting case dealing with a relatively rare primary popliteal venous aneurysm that caused relapsing episodes of pulmonary embolism in a relatively young female. During previous episodes of pulmonary embolism no triggering causes were found including screening test for malignancy and inherited coagulation disorders and remarkably Doppler venous ultrasonography was negative for thrombosis. Interestingly in the past she was found having a Becker’s cyst. And most importantly she was put on oral anticoagulation. Notwithstanding the antithrombotic medication she developed another episode of shortness of breath that turned out to be a pulmonary embolism (PE). In this context a new ultrasonographic exam of the veins of the lower limbs showed a big aneurysm of the left popliteal vein that was treated by surgical resection given its high emboligenic potential notwithstanding the antithrombotic prophylaxis. Some brief comments.
-Given the big dimension of the aneurysm (65x19x31) why was it overlooked the first time? This is a crucial point in order to understand whether you need a special skill to visualize this abnormality.
Figure 1 is not satisfying. The sac is not clearly identifiable. Moreover a compression test to document inward flow into the sac is not presented. Ideally a clip has to be added. So a better image(s) should be added or at least some arrow to delineate the sac contour has to be added.
Could there be some problem in distinguishing a becker cyst or Cysts of the proximal tibiofibu- lar joint from this popliteal venous aneurysm? It should be nice to add some comments on this point.
Why you decided to continue anticoagulation even after the aneurysmectomy? Was that a temporary decision? Please explain.
Author Response
Thank you for your suggestions, they have been immensely helpful, and we appreciate all the insightful comments on revising several aspects of our case report. Therefore, we addressed each concern and describe all the changes that we have made:
Comment 1: Given the big dimension of the aneurysm (65x19x31) why was it overlooked the first time?
Response: The patient was first investigated and treated in another service. We believe that the reason why the aneurysm was misdiagnosed was because of its localization (above the popliteal fossa) a two-point compression ultrasonography. Therefore, the two-point compression ultrasonography was negative, but a complete ultrasonography of the lower extremity venous system revealed the venous aneurysm. In order to make myself better understood I attach a mov with the compression of the popliteal vein from the popliteal fossa:
Comment 2: Figure 1 is not satisfying. The sac is not clearly identifiable. Moreover, a compression test to document inward flow into the sac is not presented. Ideally a clip has to be added. So, a better image(s) should be added or at least some arrow to delineate the sac contour has to be added.
Response: We changed Figure 1 with an image of the aneurysmal popliteal vein with partial thrombosis. I hope that this one is more satisfying. I would like to add a clip, but I don’t know if this is possible because I couldn’t find a section for uploading it.
Comment 3: Could there be some problem in distinguishing a Backer cyst or Cysts of the proximal tibiofibular joint from this popliteal venous aneurysm? It should be nice to add some comments on this point.
Response: Thank you for this suggestion. We believe that the particularities of our patient come from the rarity of popliteal venous aneurysms, but also from the rare association of these rare findings with popliteal synovial cysts, also known as Baker’s cysts. Moreover, large popliteal cysts may be responsible for compression upon the popliteal vein resulting in thrombosis. But, ultrasonography is the key investigation in differentiating between those two entities: the vein aneurysm shows sluggish, swirling blood flow inside, and the Doppler spectral analysis revealed low velocity blood flow with normal phasic variation corresponding to a venous waveform in comparison with the Baker’s cyst- a fluid-filled structure at the postero-medial knee with a 'neck' at its deepest extent, being referred in the literature as a “speech bubble”. The changes in the text appear in yellowin the revised paper.
Comment 4: Why you decided to continue anticoagulation even after the aneurysmectomy? Was that a temporary decision? Please explain.
Response: Even though definitive recommendations are hardly to be made regarding the role and duration of anticoagulation in postoperative patients because of the rarity of vein aneurysms in the clinical practice, we decided to continue anticoagulation for 3 more months due to the recent history of pulmonary embolism. We mention that this is the recommendation of anticoagulation therapy duration in patient with pulmonary embolism and a major transient/reversible risk factor, even though vein aneurysm is not mentioned in the guideline ( Konstantinides SV, Meyer G, Becattini C, Bueno H, Geersing GJ, Harjola VP, et al. 2019 ESC Guidelines for the diagnosis and management of acute pulmonary embolism developed in collaboration with the European Respiratory Society (ERS). Eur Heart J. 2020 Jan 21;41(4):543-603.).
Thank you for the valuable comments.
